# Fabrication of Yb:YAG Transparent Ceramic by Vacuum Sintering Using Monodispersed Spherical $Y_2O_3$ and $Al_2O_3$ Powders

**Jinsheng Li [1,2,*], Xin Liu [2], Lei Wu [2], Haipeng Ji [3], Liang Dong [2], Xudong Sun [1,*] and Xiwei Qi [2,4,*]**

[1] Institute of Ceramics and Powder Metallurgy, School of Materials Science and Engineering, Northeastern University, Shenyang 110819, China

[2] Key Laboratory of Dielectric and Electrolyte Functional Materials Hebei Province, School of Resources and Materials, Northeastern University at Qinhuangdao, Qinhuangdao 066004, China

[3] Key Laboratory of Advanced Energy Catalytic and Functional Material Preparation of Zhengzhou City, School of Materials Science and Engineering, Zhengzhou University, Zhengzhou 450001, China

[4] College of Metallurgy and Energy, North China of Science and Technology, Tangshan 063210, China

* Correspondence: jinsheng1986@163.com (J.L.); xdsun@mail.neu.edu.cn (X.S.); qxw@mail.neuq.edu.cn (X.Q.)

**Abstract:** In the present work, Yb:YAG (yttrium aluminum garnet) transparent ceramics were fabricated using monodispersed spherical $Y_2O_3$, $Al_2O_3$ powders and commercial $Yb_2O_3$ nano-powder as raw materials, adding 0.5 wt% tetraethyl orthosilicae (TEOS) through the solid-state method and vacuum sintering technology. The prepared monodispersed $Y_2O_3$ and $Al_2O_3$ powders adopted by homogeneous co-precipitation showed improved mixing uniformity and lead to the reduced defect of the YAG powders. After sintering in vacuum at 1700 °C for 10 h, the $(Y_{1-x}Yb_x)AG$ ($x$ = 0, 0.01, 0.10) ceramics with high transparency were obtained. Analysis of the densification rate, micromorphology, and optical properties of the ceramics suggests that the performance of the Yb:YAG ceramics is independent of the doping amount of Yb. Moreover, when the $Y_2O_3$, $Al_2O_3$, and $Yb_2O_3$ mixtures were laid aside for some time in the air after milling and drying, the performances of the as-prepared Yb:YAG ceramics would be affected distinctively. It is likely because the $Y_2O_3$ is easily hydrolyzed to $Y(OH)_3$, $Y(OH)^{2+}$ and $Y_2(OH)_2^{4+}$, which impinged the sintering activity of the powder mixture.

**Keywords:** Yb:YAG; transparent ceramics; vacuum sintering

## 1. Introduction

Yttrium aluminum garnet (YAG) belongs to the cubic crystal system and has excellent optical, mechanical properties and chemical stability. Among the solid laser materials, YAG crystal has the best comprehensive performances [1–6]. YAG doped with ($Ce^{3+}$, $Eu^{3+}$) can be widely used in the fields of scintillation and luminescent materials [7]. YAG doped with other rare earth ions ($Nd^{3+}$, $Yb^{3+}$) can be widely used as laser crystal in the fields of laser, medical instruments, and military defense. Compared with Nd:YAG, the Yb:YAG shows significant advantages, such as no concentration quenching effect of Yb doping and high quantum efficiency. As the central absorption wavelength of $Yb^{3+}$ is about 940 nm, which perfectly matches with the InGaAs diode, Yb:YAG has become the preferred solid laser gain medium with high efficiency [8,9].

Traditionally, YAG single crystal laser was grown by the Czochralski method. There are many limitations such as low growth rate, high cost and the difficulty of obtaining a large size. As the YAG transparent ceramics have similar physical, chemical and laser properties with single crystal, it is preferred to prepare YAG ceramics with low cost and a high doping concentration of rare earth ions [10,11]. Since Ikesue prepared high-quality Nd:YAG transparent ceramics and used laser diode (LD) pumping to obtain laser output in 1995, YAG transparent ceramics have developed rapidly [12].

The solid-state method and wet-chemical method were conventionally used to fabricate YAG transparent ceramics [13–21]. The wet-chemical method includes the sol–gel method, combustion synthesis method, hydroxide co-precipitation method, homogeneous precipitation method, glycol thermal treatment, spray-pyrolysis method and so on. The precursor powders contain surface water and structural water. Due to the adsorption of water, it is difficult to eliminate the hard agglomeration of the powder caused by surface tension and hydrogen bonding. Moreover, the difference in the precipitation solubility and sedimentation rate of the precursor cations in the precipitation method makes it difficult to obtain an accurate stoichiometric ratio and uniform composition of YAG powders. The solid-state method is the traditional method to prepare YAG powders during which the well-mixed high-purity oxide powders were sintered at high temperatures. This method is low cost and easy to operate and scale up. However, agglomeration could be observed in the commercial $Y_2O_3$ powders, and the particle size is usually inhomogeneous. Furthermore, the mismatch of powder particle sizes between the $Y_2O_3$ and $Al_2O_3$ powders results in poor uniform mixing. In this case, higher activation energy is required for sufficient ion diffusion, and higher calcination temperature and longer sintering period were required to obtain a YAG single phase [22,23].

In this study, Yb:YAG transparent ceramics were prepared using monodispersed spherical $Y_2O_3$ and $Al_2O_3$ powders through homogeneous coprecipitation and commercial $Yb_2O_3$ nano-powders as raw materials by solid-state process and vacuum sintering. The optical properties and micro-morphology of the YAG ceramics doped with different concentrations of Yb were analyzed. In addition, it is found that the sintering activity of the $Y_2O_3$, $Al_2O_3$ and $Yb_2O_3$ mixing powders would be greatly affected after being placed for some time after ball milling. The phase formation, micro-morphology and sintering activity of the laying aside precursor powders were also analyzed in depth.

## 2. Experimental Procedures

### 2.1. Materials

$Y_2O_3$ (99.99% purity, Huizhou Ruier Rare-Chem. Hi-Tech. Co., Ltd., Huizhou, China), $Yb_2O_3$ (99.99% purity, Huizhou Ruier Rare-Chem. Hi-Tech. Co., Ltd., Huizhou, China), $Al(NO_3)_3 \cdot 9H_2O$ (purity > 99%; Zhenxin Chemical Reagent Factory, Shanghai, China), nitric acid (analytical grade, Sinapharm, Shanghai, China), urea (analytical grade, Sinapharm, Shanghai, China) and tetraethoxysilane (TEOS, >99.99%, AlfaAesar) were used as the raw materials.

### 2.2. Preparation of Spherical $Y_2O_3$ and $Al_2O_3$ Powders

Homogeneous precipitation was used to prepare the monodispersed spherical $Y_2O_3$ powders. Firstly, $Y_2O_3$ powders were dissolved into heated $HNO_3$ at 90 °C to obtain the $Y(NO_3)_3$ solution. Then, 0.15 M $Y^{3+}$ and 0.5 M urea were mixed into 500 mL of solution and stirred equably at 90 °C for 2 h. After the reaction, the precipitate was collected via centrifugation, washed with deionized water and ethanol three times, and then dried at 80 °C for 24 h in an oven to obtain soft and white precursor powder. $Y_2O_3$ powders were obtained by grinding and sieving the precursor powder and then calcination in a muffle furnace at 900 °C for 4 h. The main chemical reactions for the preparation of $Y_2O_3$ are as follows:

$$y\text{Y}^{3+} + x\text{NH}_4^+ + z\text{OH}^- + m\text{CO}_3^{2-} + n\text{H}_2\text{O} \leftrightarrow (\text{NH}_4)_x\text{Y}_y(\text{OH})_z(\text{CO}_3)_m \cdot n\text{H}_2\text{O} \quad (1)$$

$$(\text{NH}_4)_x\text{Y}_y(\text{OH})_z(\text{CO}_3)_m \cdot n\text{H}_2\text{O} \leftrightarrow x\text{NH}_3 + 0.5y\text{Y}_2\text{O}_3 + m\text{CO}_2 + n\text{H}_2\text{O} \quad (2)$$

The process to obtain the monodispersed spherical $Al_2O_3$ is similar to that of $Y_2O_3$. Firstly, $Al(NO_3)_3 \cdot 9H_2O$ is dissolved in water to obtain a certain concentration of $Al(NO_3)_3$ solution. Then, 0.006 M $Al(NO_3)_3$ and 0.005 M $(NH_4)_2SO_4$ and 0.12 M urea were mixed to form 500 mL solution and stirred equably at 90 °C for 1.5 h. After that, the precipitate was collected via centrifugation, washed with deionized water and ethanol, and then dried at

80 °C for 24 h in an oven to obtain soft and white precursor powder. $Al_2O_3$ powders were obtained by grinding, sieving and calcinating the precursor powder in a muffle furnace at 1000 °C for 4 h. The main chemical reactions for the preparation of $Al_2O_3$ are as follows:

$$yAl^{3+} + xNH_4^+ + zOH^- + mCO_3^{2-} + nH_2O \leftrightarrow (NH_4)_xAl_y(OH)_z(CO_3)_m \cdot nH_2O \quad (3)$$

$$(NH_4)_xAl_y(OH)_z(CO_3)_m \cdot nH_2O \leftrightarrow xNH_3 + 0.5yAl_2O_3 + mCO_2 + nH_2O \quad (4)$$

### 2.3. Preparation of Yb:YAG Transparent Ceramics

The Yb:YAG transparent ceramics were prepared by the solid-state reaction method and vacuum sintering technique. The prepared monodispersed spherical $Y_2O_3$, $Al_2O_3$ powders, and commercial $Yb_2O_3$ nano-powders were weighed according to the stoichiometric ratio. The Yb doping concentrations were 0%, 1 at % and 10 at%, respectively. Tetraethyl orthosilicate (TEOS, 0.5 wt%) was used as the sintering aids. Ball milling was then used to mix the powders uniformly. Absolute ethyl alcohol was used as the ball-milling media, and the ratio of solid and liquid was 1:4. The mixture was milled with corundum grinding ball for 16 h and dried in an oven at 80 °C for 24 h. The mixture powder was calcined in air for 1200 °C, 1300 °C, 1400 °C and 1500 °C to analyze the formation of the YAG phase.

$(Y_{1-x}Yb_x)AG$ powders prepared were formed by the bidirectional pressing of a steel mold and cold isostatic pressing at 200 MPa to obtain green products with a thickness of 5 mm and a diameter of 13 mm. The green products were then sintered at 1600–1700 °C in a vacuum furnace with the heating and cooling rates of 5 °C/min and 10 °C/min, respectively. Then, the ceramic products after sintering were annealed in the air at 1300 °C to eliminate the oxygen vacancies. Finally, the ceramic products were polished to measure the transmittance and observe the microstructure.

### 2.4. Characterization

Phase composition analysis of the samples was performed by X-ray diffraction (XRD, Model PW3040/60; PANALYTICAL B.V, Amsterdam, The Netherlands) under 40 kV/40 mA using nickel-filtered Cu K$\alpha$ radiation at a scanning speed of 4.0°·$2\theta$/min and a step size of 0.02°. The morphologies were observed using field emission scanning electron microscopy (Model JSM-7001F; JEOL, Tokyo, Japan) with an acceleration voltage of 10 kV. The densification behaviors of YAG powder compacts in flowing $O_2$ were analyzed through dilatometry by a thermal mechanical analyzer (SETSYS Evolution 1750; Setaram, Lyons, France) at a constant heating rate of 10 °C/min and a cooling rate of 15 °C/min. The in-line transmission of the ceramic samples was measured by an ultraviolet-visible spectrometer (PerkinElmer Lambda750S, Waltham, MA, USA).

## 3. Results and Discussion

Figure 1a,b show the morphologies of the as-prepared $Y_2O_3$ precursor and that calcined at 900 °C. Both precursor powders were monodispersed and uniform. The average particle size of precursor was about 400 nm, and that of calcined powders was approximately 200 nm. Figure 1c shows the morphology of the commercial $Yb_2O_3$ nano-powder. The average particle size was about 40 nm.

Figure 2 shows the morphologies of as-prepared $Al_2O_3$ precursor and that calcined at 1000 °C. Both were monodispersed and homogeneous. Shrinkage of the precursor powder was obvious, as the size decreased from 700 nm to about 400 nm. This is due to the volatility of $NH_4^+$, $CO_3^{2-}$ and $OH^-$ in the precursor during calcination.

Figure 3 exhibits the XRD patterns of the products of the YAG precursor sintered at different temperatures. The powders calcined at 1200 °C contain phases of YAG ($Y_3Al_5O_{12}$, JCPDS card No. 33-0040), YAP ($YAlO_3$, JCPDS card No. 70-1677), YAM ($Y_4Al_2O_9$, JCPDS card No. 14-0475), and α-$Al_2O_3$. The dominant phase was YAP. When calcined at 1300 °C, the YAM phase was completely transformed into the YAP phase. At 1400 °C, only a small amount of YAP existed as the impurity phase, and the major phase was YAG. At 1500 °C,

only the YAG phase remained. The phase evolution is quite similar to that previously reported [24]. After calculation, the lattice parameter of the $(Y_{1-x}Yb_x)AG$ ($x$ = 0, 0.01, 0.10) crystals calcined at 1500 °C are 1.2002 nm, 1.1986 nm and 1.1964 nm, respectively, because the ionic radius of Yb is smaller than that of the Y ionic radius. After Yb entered the YAG lattice, it will lead to lattice contraction. So, the lattice constant decreased with the increase in Yb doping.

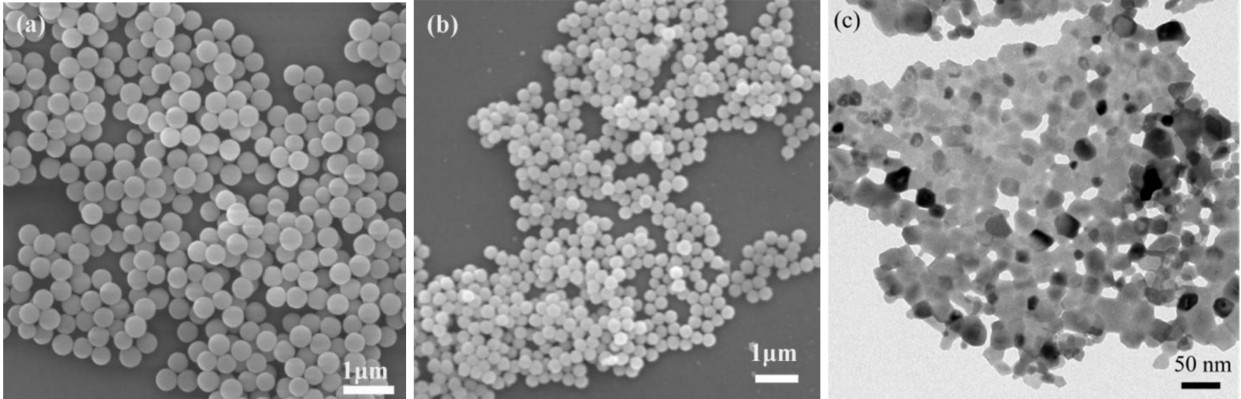

**Figure 1.** Micrographs showing particle morphologies of the $Y_2O_3$ precursor (**a**) and calcined at 900 °C (**b**) and commercial $Yb_2O_3$ nano-powder (**c**).

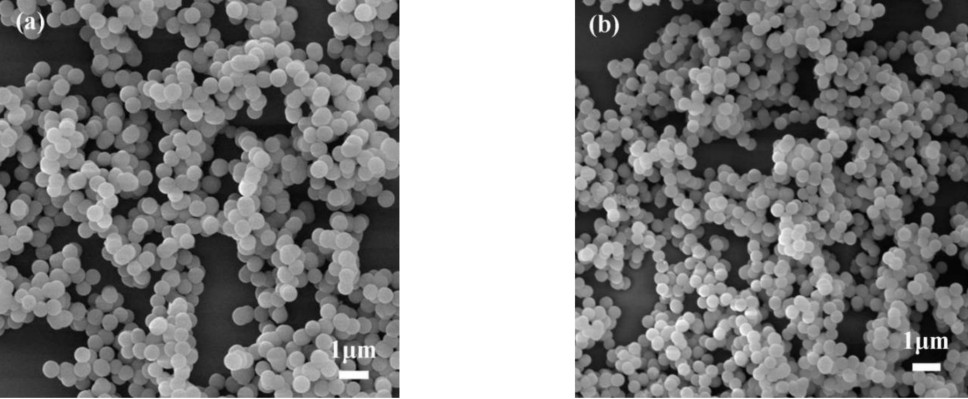

**Figure 2.** Micrographs showing morphologies of the $Al_2O_3$ precursor (**a**) and calcined at 1000 °C (**b**).

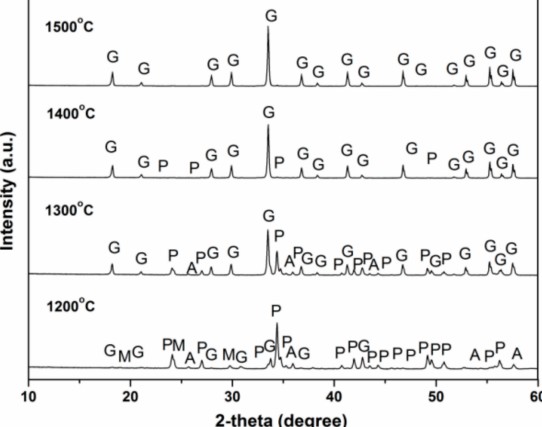

**Figure 3.** XRD patterns of the products calcined at various temperatures. G represents YAG garnet, P represents YAP phase, M represents YAM phase, and A represents $Al_2O_3$ phase.

For the formation of the YAG phase, the diffusion of ions during the solid-state reaction needs to overcome the energy barrier. The relation between the diffusion coefficient ($D$) and temperature ($T$) can be expressed by Equation (5) [25]:

$$D = D_0 \exp\left(\frac{-Q}{RT}\right) \qquad (5)$$

where $D$ is the self-diffusion coefficient, $D_0$ is the pre-exponential factor, $Q$ refers to the self-diffusion activation energy, $R$ is the Boltzmann constant, and $T$ is the thermodynamic temperature. The exponential term ($-Q/RT$) is the probability that an atom overcomes the energy barrier transition.

According to Equation (5), the atomic diffusion coefficient is exponentially related to temperature. When the temperature rises, the diffusion coefficient increases as well. Thus, high temperature is required to enhance atomic diffusion and obtain the YAG phase.

In addition, particle size is a vital parameter that influences the reaction rate in a solid-state reaction. The particle size can change the reaction interface, diffusion cross-section and particle surface structure. Equation (6) was used to describe the relationship between the rate constant and particle size of reactants in a solid-state reaction [26].

$$K_f = \frac{2DC_0}{r^2} \qquad (6)$$

where $K_f$ is the reaction rate constant, $C_0$ is the concentration of the reactant, $D$ is the atomic diffusion coefficient, and $r$ refers to the particle radius of the reactant.

As shown in Equation (6), the rate of solid reaction increased with the reduction in particle radius. The smaller the particle size, the larger the specific surface, the larger the reaction cross-section, the flatter the distribution curvature of bond strength, the larger the proportion of weak bonds, and the greater the reaction and diffusion ability. The temperature required for the YAG phase formation starting from the monodispersed $Al_2O_3$ spherical powders was higher than that using the $Al_2O_3$ nano-powders in the previous work under the same conditions, which was mainly due to the low activity and slow diffusion rate of large spherical $Al_2O_3$ particle size in the present work [27].

The mixing uniformity of oxide powders also matters. When the raw materials failed to be mixed thoroughly, diffusion would be difficult because of a long distance between cations. Therefore, a higher temperature was needed to eliminate the transient phase. Meanwhile, when hard agglomeration appeared in a raw mixture, the activity of the powders would be poor, which would lead to an uneven mixture. In this experiment, monodispersed spherical $Y_2O_3$ and $Al_2O_3$ powders improved the mixing uniformity, which then greatly reduced the defect of final ceramics.

Figure 4 exhibits the shrinkage and shrinkage rate curves during sintering of the $(Y_{1-x}Yb_x)AG$ ($x$ = 0, 0.01, 0.10) products at a constant heating rate (10 °C/min). The expansion phenomenon can be observed in the samples from 400 to 1200 °C, which can be explained by the phenomenon of expansion on heating and contraction on cooling. Linear shrinkage and linear shrinkage rate were similar, which are independent on the doping concentration of Yb. The shrinkage was about 8%, and the maximum shrinkage rate was about 0.8 %/min.

Sharp contraction can be observed around 1250 °C, which can be attributed to the difference in density that came from the YAM to YAP transformation process. The result is consistent with the XRD analysis. Furthermore, the densification process is another reason for the contraction. The expansion phenomenon can be observed when the temperature reached about 1450 °C, which was mainly due to the YAP to YAG transformation process. However, the densification rate was lower than the expansion rate coming from the phase transition. When the temperature was higher than 1500 °C, the densification led to a constant sample volume shrinking.

Figure 5 shows the thermally etched surface morphologies of $(Y_{1-x}Yb_x)AG$ ceramic samples sintered at 1600 °C. Similar surface morphologies and the average particle size of

the samples can be observed. The particle size was about 1.2 μm, and the relative density of the Yb:YAG ceramic sample was about 97.5%. The pore distribution was uniform and the pores were basically distributed at particle boundaries. There were no large pores, which was beneficial for the late stage of sintering. At the evaluated sintering temperature, the small pores could be easily eliminated along grain boundaries with the grain growth.

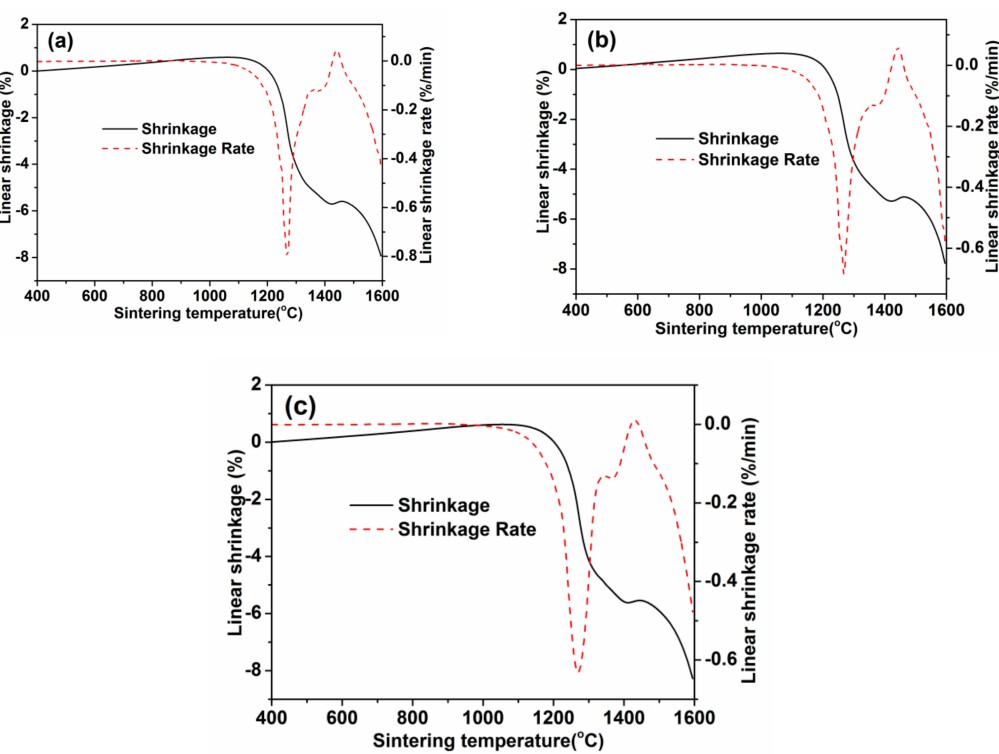

**Figure 4.** Shrinkage and shrinkage rate during constant heating rate (10 °C/min) sintering of the $(Y_{1-x}Yb_x)AG$ ($x$ = 0, 0.01, 0.10) compacts, (**a**) $x$ = 0, (**b**) $x$ = 0.01, (**c**) $x$ = 0.1.

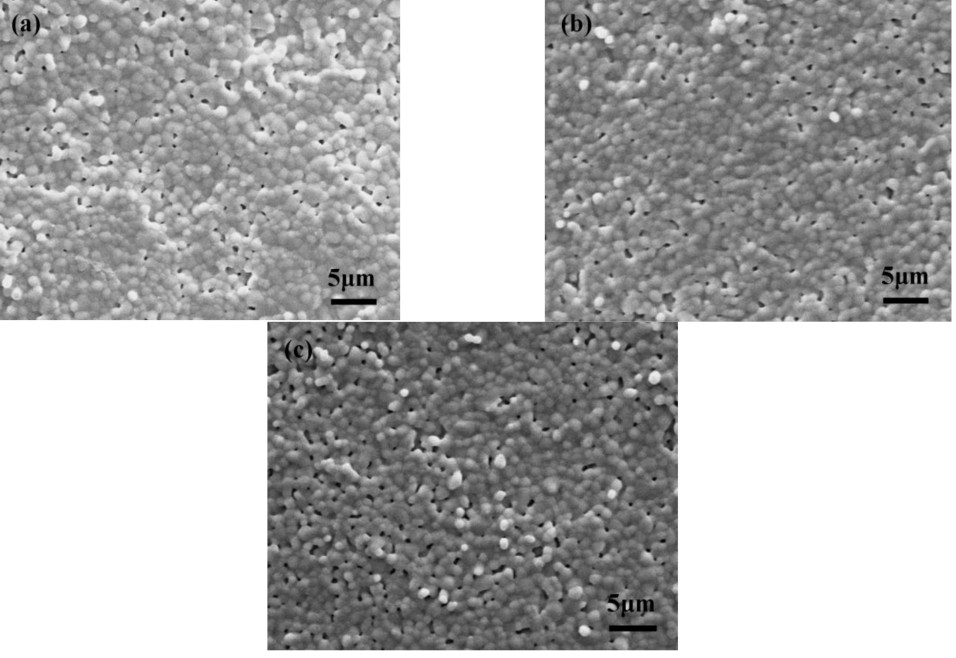

**Figure 5.** Thermally etched surface morphologies of $(Y_{1-x}Yb_x)AG$ ceramic samples sintered at 1600 °C, (**a**) $x$ = 0, (**b**) $x$ = 0.01, (**c**) $x$ = 0.1.

　　　　Figure 6 shows the thermally etched surface morphologies of the Yb:YAG transparent ceramics sintered at 1700 °C. Compared with those sintered at 1600 °C, the grain size increased distinctly, and almost all pores were removed. Grain boundary diffusion and volume diffusion were the main mechanisms when sintered at 1700 °C. In the later stage of sintering, pores were completely isolated. Then, a sum effect of grain growth, hole coarsening, and volume shrinkage finally resulted in an actual density of the ceramic that was close to the theoretical density. The average particle size of the ceramic under the vacuum sintering at 1700 °C was about 5.5 μm using the Win Roof statistics. The relative density of the Yb:YAG ceramic sample was about 99.8%.

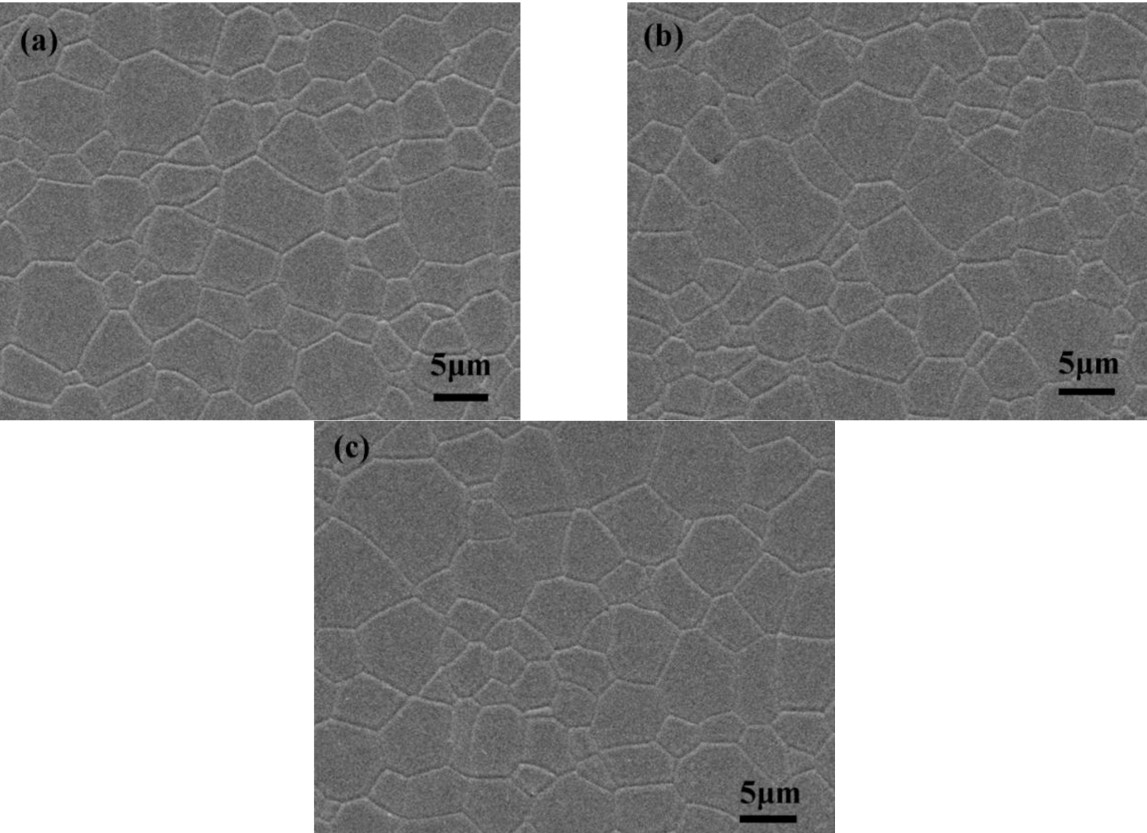

**Figure 6.** Thermally etched surface morphologies of $(Y_{1-x}Yb_x)$AG transparent ceramics sintered at 1700 °C, (**a**) $x$ = 0, (**b**) $x$ = 0.01, (**c**) $x$ = 0.1.

　　　　As mentioned above, in spite of the change of the $Yb^{3+}$ content, the densification rate and surface microscopic morphologies of the Yb:YAG ceramics were similar, which is consistent with previous study [27]. It is thus proved that the doping of Yb had little influence on the pores discharge, grain growth and densification of ceramics.

　　　　Figure 7 shows the EDX analysis of the $(Y_{0.94}Yb_{0.06})$AG transparent ceramic sintered at 1700 °C. The Y, Yb, Al, Si and O elements were uniformly distributed, and there was no concentration gradient and segregation. When sintered at high temperatures, the addition of Si formed the Y-Si-O liquid phase. The existence of the liquid phase made the YAG ceramics more conducive to particle rearrangement, dissolution–precipitation, and densification in the middle and late stages of sintering. The solubility of $SiO_2$ on the surface of the YAG particles was restricted. When the liquid phase exceeds the solubility on the particle surface, a Si-rich glass phase is easily formed [28]. The addition of Si content was 0.5 wt % in our study, which was less than its solubility on the YAG grain surface. Such a small amount of Si doping will promote densification but will not become enriched along the grain boundaries.

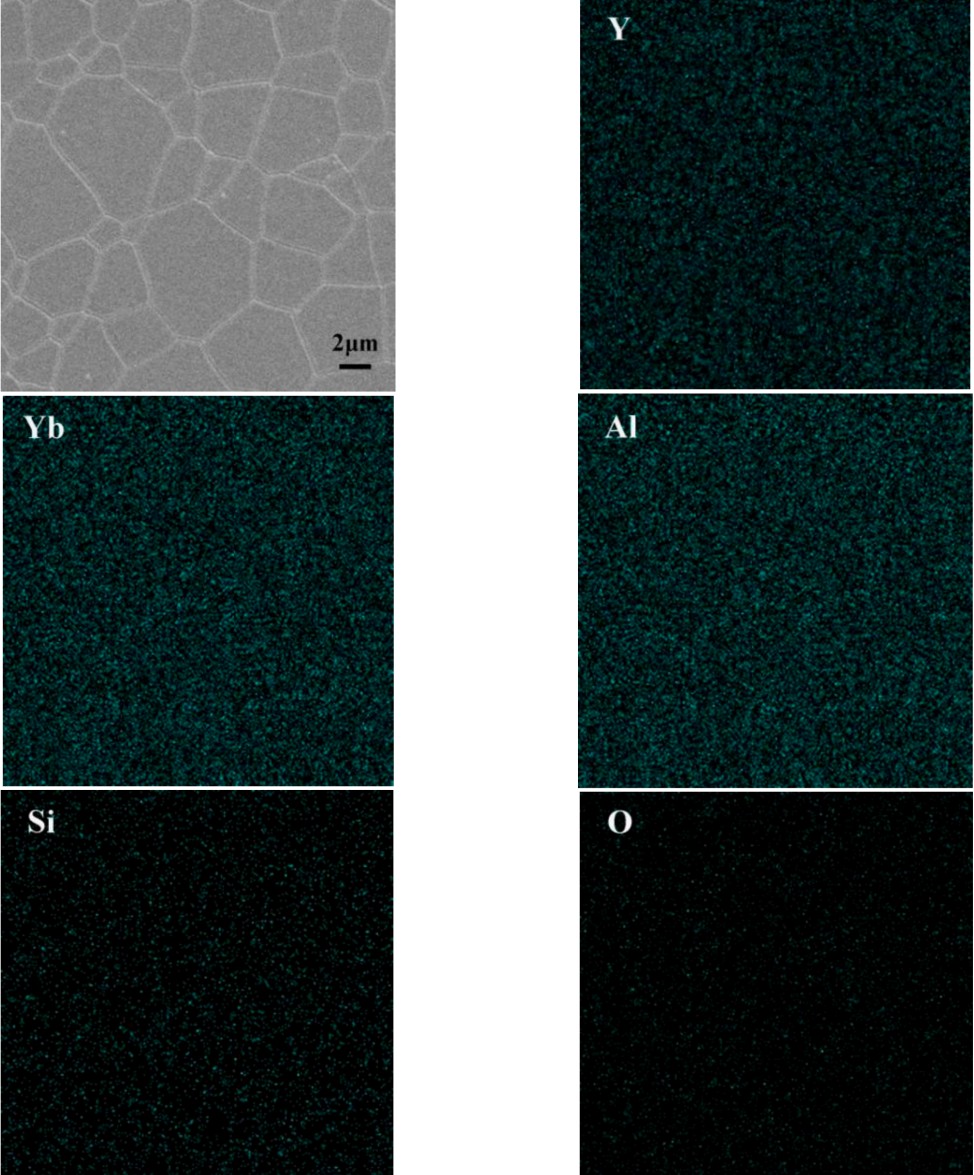

**Figure 7.** EDX spectra analysis of the composition distributions of the $(Y_{0.94}Yb_{0.06})AG$ transparent ceramic sintered at 1700 °C.

Unlike $Nd^{3+}$, the ionic radius of $Yb^{3+}$ was small, and it is thus easier for $Yb^{3+}$ to enter the YAG lattice. In addition, because of the low segregation coefficient of $Yb^{3+}$ (0.18), there is a limited lattice drag effect when $Yb^{3+}$ was doped in YAG. The doping of Yb did not promote or inhibit the densification process of the YAG transparent ceramics [29–33].

$(Y_{1−x}Yb_x)AG$ transparent ceramics were obtained by dry pressing and isostatic pressing, which were sintered in vacuum at 1700 °C for 10 h. Figure 8 shows the $(Y_{1−x}Yb_x)AG$ transparent ceramics (1.2 mm thick, both sides polished) with different doping contents of Yb (from left to right: $x = 0$, 0.01 and 0.10, before annealing and $x = 0$, 0.01 and 0.10, after annealing). All the samples were observed to have fine optical transmittance.

Figure 9 shows the optical transmittance of the Yb:YAG transparent ceramics. There were four absorption peaks at 250 nm, 300 nm, 380 nm and 640 nm before annealing. The ytterbium mainly existed in the form of $Yb^{2+}$ in the reduction atmosphere. The absorption peaks at 250 nm and 300 nm are due to the defects (oxygen vacancies) produced by fast neutrons ion irradiation via elastic collision, and they form F and $F^+$ center. The absorption peak at 300 nm is weak. The optical absorption spectrum increases with fluence up to

saturation level. So, sometimes, the 300 nm absorption peak does not appear [34]. The absorption peak at 380 nm was attributed to the absorption of $Yb^{2+}$ ions, and the peak at 640 nm was due to the oxygen vacancies that existed in the ceramics sintered in vacuum. The ceramics had a slightly higher linear transmittance after annealing due to the removal of oxygen vacancies and color centers after annealing.

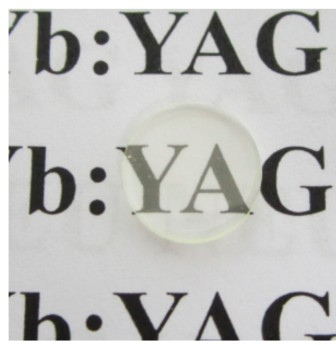 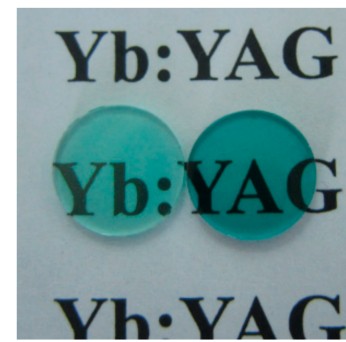 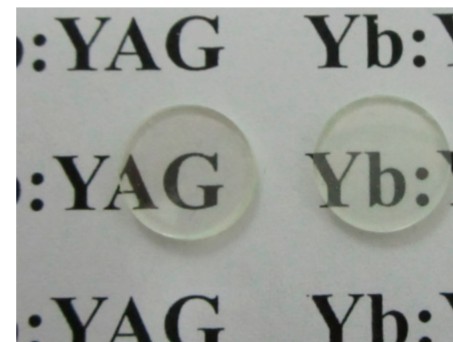

**Figure 8.** Photos of $(Y_{1-x}Yb_x)AG$ transparent ceramics (1.2 mm thick), from left to right: $x = 0$, 0.01 and 0.1, before annealing and $x = 0$, 0.01 and 0.1, after annealing.

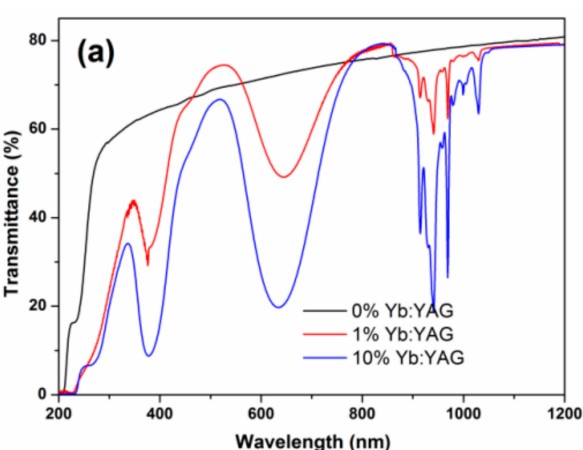 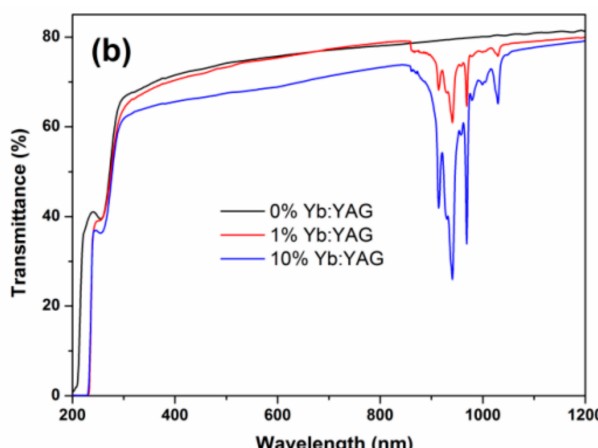

**Figure 9.** Optical transmittance of $(Y_{1-x}Yb_x)AG$ ($x = 0$, 0.01 and 0.1) transparent ceramics, (**a**) before annealing, and (**b**) after annealing.

The optical transmittance of the samples after annealing were 81.43%, 79.93% and 79.15%, respectively, at 1200 nm with the increasing of Yb contents. The in-line transmission of the sintered Yb:YAG samples had high transmission and was comparable to YAG single crystal grown by the Czochralski method (about 84% transmission). The minimum optical transmittance can be seen at 916 nm, 941 nm and 968 nm, which can be explained by the absorption of light in Yb:YAG ceramics. This matches with the transition of $Yb^{3+}$ from the $^2F_{7/2}$ ground state to the $^2F_{5/2}$ excited state.

According to the mathematical expression of Beer–Lambert law, A = lg(1/T) = Kbc. T is the transmission ratio (light transmittance), and B is the absorption layer thickness. Obviously, as with Er:YAG transparent ceramic, the absorption band of Yb:YAG transparent ceramic does not satisfy Beer's law. The factors affecting the transmittance of YAG ceramic include pores, impurities, defects and grain boundaries, which have nothing to do with the ceramic thickness.

It is worth mentioning that the sintering activity of the $Y_2O_3$ and $Al_2O_3$ powders was apparently influenced if the precursor powders were laid aside for a period in the air after milling and drying. The precursor powders and ceramic sample performances were studied using monodispersed spherical $Y_2O_3$, commercial $Al_2O_3$ and $Yb_2O_3$ powders through the solid-state method and the vacuum-sintering technique.

Figure 10 shows the morphologies of the $(Y_{0.94}Yb_{0.06})$AG precursor calcined at 1100 °C. It can be seen that agglomeration can be observed in $(Y_{0.94}Yb_{0.06})$AG precursor after laying aside for a month, which was unfavorable as the raw materials for preparing transparent ceramics. Powder agglomeration has always been an obstacle for transparent ceramics sintering. This is because during the sintering process, the agglomerated grains would grow abnormally, and the pores would be retained in the grains.

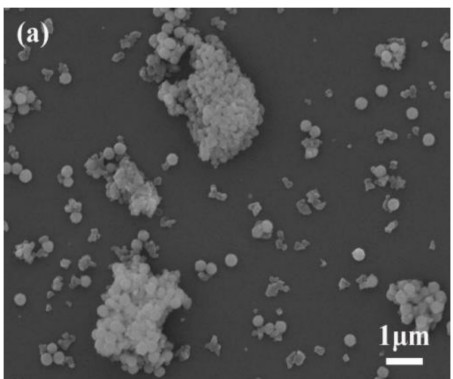 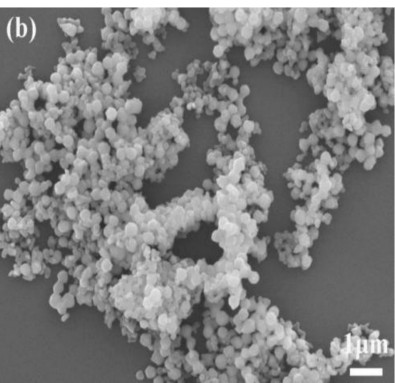

**Figure 10.** Micrographs showing particle morphologies of the precursor of $(Y_{0.94}Yb_{0.06})$AG powders laying aside a month (**a**) and the precursor calcined at 1100 °C (**b**).

Figure 11 exhibits the ceramics obtained with different precursor powders via vacuum sintering at 1700 °C for 5 h. It can be seen that the ceramics obtained using fresh precursor were transparent and the relative density was 99.98% after calculation. In comparison, the ceramic obtained with the precursor placed in air for a month was opaque, and the relative density was only 97.87%.

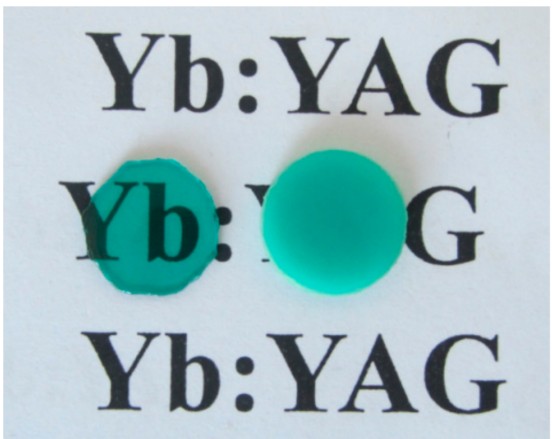

**Figure 11.** Photos of $(Y_{0.94}Yb_{0.06})$AG transparent ceramics calcined at 1700 °C, 5 h, the left: the ceramic of the precursor directly sintering; the right: the ceramic of the precursor laying aside a month and sintering at the same condition.

Figure 12 shows the thermally etched surface morphologies of $(Y_{1-x}Yb_x)$AG ceramics sintered at 1700 °C. The surface of the ceramics shown in Figure 12a was clean, and there were no pores using the fresh precursor powders. Meanwhile, as observed in Figure 12b, there were large poles on the surface of ceramics when using the precursor powder after laying aside for a month. Poles acted as the scattering centers of light and thus had a remarkable influence on the optical properties of the ceramics. According to Greskovich's report, as the enclosed pores rose from 0.25% to 0.85%, the optical transmittance decreased by 33% [35].

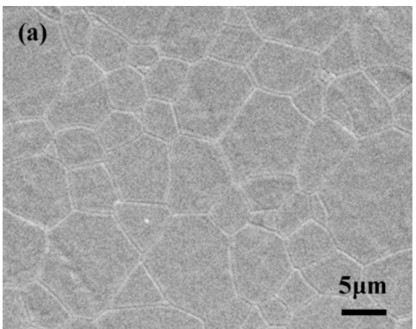 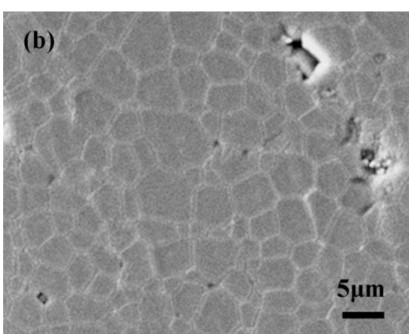

**Figure 12.** Thermally etched surface morphologies of $(Y_{0.94}Yb_{0.06})AG$ ceramics sintered at 1700 °C, (**a**) the precursor not laying aside, (**b**) the precursor laying aside a month.

　　　Figure 13 shows the XRD patterns of the ceramics after vacuum sintering at 1700 °C. It can be observed that both ceramics consist of a single YAG phase. Thus, the main reason for the decreased transmittance for the Yb:YAG ceramic obtained from the stale precursor is the relatively high porosity rather than the influence of the second phase. The only difference between the two types of precursor powders was the storage time in air. The results were closely related to the agglomeration of precursor powders. The sintering activity of stale precursor powders became lower and impacted the optical transmittance of YAG ceramics.

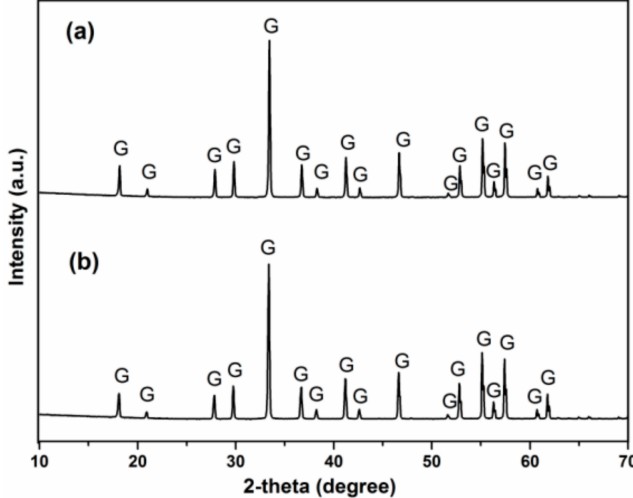

**Figure 13.** XRD patterns of the ceramics after vacuum sintering: (**a**) the precursor laying aside a month, compaction and sintering, (**b**) the precursor powers immediately compaction and sintering.

　　　Based on the previous studies [36–41], the main dissolved ions of $Y_2O_3$ are not only $Y^{3+}$ but also $Y(OH)^{2+}$ and $Y_2(OH)_2^{4+}$, which are positive polyvalent ions and easily adsorbed to the surface of negatively charged materials. In addition, the $Y_2O_3$ powders were easy to absorb $H_2O$ and $CO_2$ in the air, which changes the surface activity of powders. The reaction process can be expressed as follows:

$$Y_2O_3 + H_2O \rightarrow Y_2O_3 \cdot H_2O \rightarrow Y(OH)_3 \rightarrow Y^{3+} + 3OH^- \tag{7}$$

$$CO_2 + 2OH^- = CO_3^{2-} + H_2O \tag{8}$$

$$2Y^{3+} + 3CO_3^{2-} + nH_2O = (Y_2CO_3)_3 \cdot nH_2O \tag{9}$$

　　　The XRD pattern of the precursor leftover after a month is shown in Figure 14. In addition to the diffraction peaks of $Y_2O_3$ and $Al_2O_3$, there were also impurity peaks of $C_3H_3O_6Y \cdot 2H_2O$ (i.e, $Y(HCOO)_3 \cdot 2H_2O$) induced by the interaction between the precursor powders, the added organic active agent, $H_2O$, and $CO_2$ in the air.

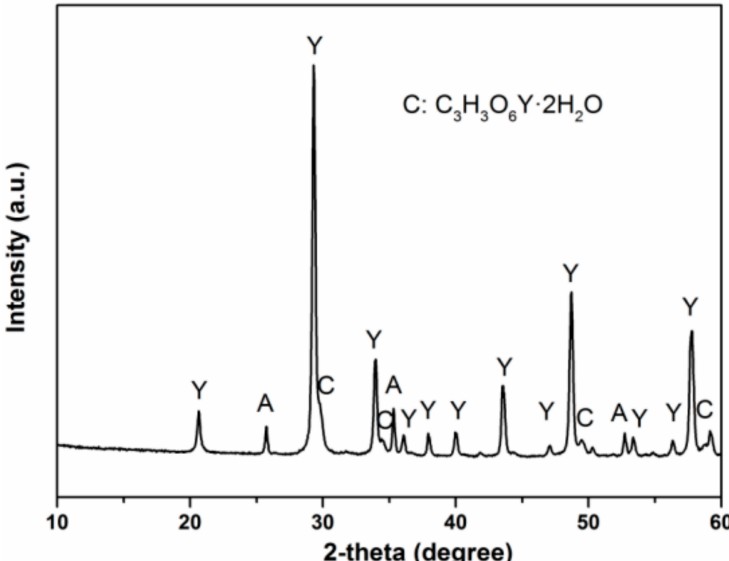

**Figure 14.** XRD pattern of the precursor powders after a month.

The precursor agglomeration was mainly due to the high specific surface area of the raw materials and the weak physical force of the powders, which includes the van der Waals' force, electrostatic interaction, capillary tension and magnetic force. The action range of van der Waals force is less than 100 μm, and the van der Waals force has a strong effect on the powder particles whose size is less than 0.05 μm. During calcination, a sintering bond occurred among small particles and led to the powder agglomeration. Another reason for powders agglomeration was that the vapor pressure of the grain surface is high. When leftover in the air, the surface of the powder would be wetted by air, which was related to the humidity of air and the curvature of the particles. At the contact point of the powder particles, capillary bridging was built by wetting solution, and the particles became agglomerated. The attraction force ($F$) among the particles has little relationship with the wetting solution but is mainly influenced by gas–liquid interface energy $\gamma_{LV}$ and particle diameter $D$. The relational expression of these factors is shown in Equation (10) [42]:

$$F = 5\gamma_{LV}D \tag{10}$$

The mass of the particle depends on the size of the particle. So, the ratio of the aggregate force to the mass is large for small particles. The agglomeration intensity $\sigma$ can be expressed by Equation (11):

$$\sigma = \frac{7S\gamma_{LV}(1-\theta)}{D\theta} \tag{11}$$

where $S$ is saturation of the wetting liquid, $\theta$ is the degree of porosity and $D$ refers to particle diameter. As shown in Equation (11), capillary tension has an obvious effect on the powder particle size under 100 μm.

According to the investigation above, after milling, sieving, and laying aside in the air for a period, $Y_2O_3$ powders can easily react with the addition of an active agent and $H_2O$ and $CO_2$ in the air. Therefore, the powder surface characteristics had been influenced. After calcination, $Y_2O_3$ powders were partly agglomerated, which impacted the course of reaction with $Al_2O_3$ powders. During the high-temperature vacuum sintering, as the surface energy of the agglomeration powders is small, the aggregated powders were preferential sintered. The densification rates of the different parts of the sintered body were different. Therefore, it was difficult for pores to exclude along grain boundaries. The existence of pores in the gain particles made the ceramics opaque.

## 4. Conclusions

Yb:YAG transparent ceramics were fabricated using commercial $Yb_2O_3$ nano-powders and the prepared monodispersed spherical $Y_2O_3$ and $Al_2O_3$ powders as raw materials by vacuum sintering.

(1) The homogeneous co-precipitation method was adopted for the preparation of spherical $Y_2O_3$ and $Al_2O_3$ powders, and the average particle size was about 200 nm and 400 nm, respectively. The prepared spherical powders improved the mixing uniformity and greatly reduced the defect in the solid-state method of preparing YAG ceramics.

(2) Yb:YAG transparent ceramics were fabricated using vacuum sintering at 1700 °C for 10 h. The optical transmittance of the ceramics can achieve up to 80% at 1200 nm. From the analysis of densification rate, micromorphology and optical properties of the ceramics, it was found that the performances of the Yb:YAG ceramics are independent of the doping amount of Yb.

(3) The $Y_2O_3$, $Al_2O_3$ and $Yb_2O_3$ mixing precursors were laid aside for a period of time in the air after milling and drying, and the sintering activity of the powders and the obtained Yb:YAG ceramic products would be influenced significantly due to the fact that the yttrium oxide was easily hydrolyzed and it was facile to react with the active agent and air.

**Author Contributions:** Methodology, X.S.; validation, X.Q., J.L. and X.S.; formal analysis, X.L., L.D., H.J. and L.W.; data curation, L.W.; writing—original draft preparation, J.L.; writing—review and editing, J.L. All authors have read and agreed to the published version of the manuscript.

**Funding:** This work was supported by the S&T Program of Hebei (22567627H) and the Natural Science Foundation of Hebei Province (Grant No. E2018501042 and E2021501017) and the Young Talents Program of Hebei Province (Grant No. BJ2020202) and the Hebei Key Laboratory of Dielectric and Electrolyte Functional Material, Northeastern University at Qinhuangdao (HKDEFM2021301).

**Institutional Review Board Statement:** Not applicable.

**Informed Consent Statement:** Not applicable.

**Data Availability Statement:** Not applicable.

**Conflicts of Interest:** The authors declare no conflict of interest.

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
