# Peer review of "Fabrication of Yb:YAG Transparent Ceramic by Vacuum Sintering Using Monodispersed Spherical Y2O3 and Al2O3 Powders"

_coatings, doi:10.3390/coatings12081155_

Round 1

Reviewer 1 Report

This is a decent article that can be recommended for publication after some improvement in the text.

1.     In the introduction, to attract a wider readership it is necessary to note other applications of YAG.

2.     The same could important for short description of different preparation methods, especially when we consider transparent ceramics.

3.     Is it possible to compare the obtained transmission spectra with similar ones for single crystals.

4.     What explains the structure of the transmission spectra in the short-wavelength region of the spectrum ? This is very similar to the so-called F-type centers, or oxygen vacancies with trapped electrons, which can form both thermally or under irradiation.  See, for example,

Izerrouken M., Meftah A., Nekkab M. Radiation damage induced by swift heavy ions and reactor neutrons in Y3Al5O12 single crystals. (2007) Nuclear Instruments and Methods in Physics Research, Section B: Beam Interactions with Materials and Atoms, 258 (2) , pp. 395-402.

Mironova-Ulmane N., Sildos I., Vasil'chenko E., et al Optical absorption and Raman studies of neutron-irradiated Gd3Ga5O12 single crystals. (2018) Nuclear Instruments and Methods in Physics Research, Section B: Beam Interactions with Materials and Atoms, 435 , pp. 306-312.

Reviewer 2 Report

Referee report on “Fabrication of Yb:YAG transparent ceramic by vacuum sintering using monodispersed spherical Y2O3 and Al2O3 powders” by Jinsheng Li et al.

This is a good article that might be suitable for publication after taking into account the following suggestion, which will certainly improve this manuscript.

1.     Introduction.  Line 30. Somewhere here, it is important to mention that YAG, in addition to laser applications, is a very important scintillation and luminescent material. See, few recent paper and references therein:

Polisadova, E.; et al. Opt. Mater. 201996, 109289. 

2.     Line 40 and 42. The end of the sentences needs supporting references.

3.     Line 46. In recent years, few more methods, such as electron-beam assisted methods was also introduced.  See, for example, Karipbayev, Z.T.; et al. Nucl. Instrum. Methods Phys. Res. B. Beam Interact. Mater. Atoms 2020479, 222–228.

4.     Lines 47 -59. Some sentences need supporting references.

5.     Line 147. Could you compare this Q value with literature data for YAG.

6.     Figure 9. While, all features of the transmission spectra in the region of 350-1200 nm were explained, the structure in the region of 200-300 nm remained unexplained. However, according Table 2, in review paper (Popov, A.I.; Kotomin, E.A.; Maier, J. Basic properties of the F-type centers in halides, oxides and perovskites. Nucl. Instrum. Methods Phys. Res. Sect. B 2010, 268, 3084–3089) such structure in the region of 200–300 nm can apparently be explained as an optical manifestation of the formation of oxygen vacancies

7.     It will be important to compare the optical absorption tail of the produced ceramics with that for single crystals.

Reviewer 3 Report

Please underline the novelty of your study as compared with the numerous literature on the same subject.

From the text of the manuscript, it is not clear why the authors used commercial Yb2O3 nano-powder but decided to prepare Y2O3 and Al2O3 powders. In other words, what is the difference between commercial Y2O3, Al2O3, and Yb2O3 powders? Why the commercial Yb2O3 was good for ceramic preparation while the other oxides were not good.

Line 16. “The prepared monodispersed Y2O3 and Al2O3 powders adopted by homogeneous co-precipitation showed improved mixing uniformity and lead to the reduced defect of the YAG powders.” The authors do not compare the YAG powder prepared using commercial and specially prepared Y2O3 and Al2O3 powders. Therefore it is not clear, how the improved mixing uniformity and reduced defect of the YAG powders were demonstrated.

Please mention in Abstract that you used TEOS as a sintering aid.

The preparation section should have more details. Please provide t chemical reactions for the preparation of Y2O3 and Al2O3 powders. Please mention temperature of the heated HNO3 (line 78). Please indicate calcination temperatures (lines 83 – 84 and 91).

The authors present the micrographs of the Y2O3 and Al2O3. Please provide the micrograph of commercial Yb2O3 nano-powder.

From Fig. 3 becomes clear that YAG precursor were sintered at different temperatures. This procedure was not described in the preparation section. Please make correction.

Line 136. Please provide the numbers of standard XRD cards that you used for the identification of the crystalline phases.

The meaning of the sentence presented in lines 160 – 163 is not clear and requires a more detailed explanation.

In Fig. 7, please give information about the colors which you used for the designation of different ions.

Line 226. The authors state that “it is thus easier for Yb3+ to enter the YAG lattice.” Please estimate the lattice parameter of Yb-doped YAG crystals for x = 0, 0.01 and 0.10 and discuss its change with Yb concentration, if any.

Please discuss the origin of absorption bands in the visible spectral range in spectra of ceramics before their annealing.

Please check if the Beer’s law is fulfilled for the absorption band of Er3+.

Below please see the technical comments.

Line 15. Please write “commercial Yb2O3 nano-powder” instead of “commercial Yb2O3 nano-powder”.

Line 32. Please change “elements” to “ions”.

Line 36. Please explain what instrument or device you mean by “InGaAs”.

Line 42. Please add the word ”ions” to the words “rare earth”.

Line 56. Please delete the word “two”.

Line 149. Please correct the sentence “The exponential term is the probability of atoms that can jump over an energy barrier”.

Line 153. Please correct typo.

Line 300. Please correct typo.

Lines 306 - 308. Please correct the typo in the term “van der Waals force”.

Lines 324 – 325. Please make the grammar correction of the sentence.

Please correct the author names in ref. [3]. They should be written as F.F. Malyavina, V.A. Taralaa, S.V. Kuznetsovb, A.A. Kravtsova, I.S. Chikulinaa, M.S. Shamaa, E.V. Medyanika, V.S. Ziryanova, E.A. Evtushenkoa, D.S. Vakalova, V.A. Lapina, D.S. Kuleshova, L.V. Taralaa, L.M. Mitrofanenko.

Please correct the author names in ref. [4]. They should be written as A. Katz, E. Barraud, S. Lemonnier, E. Sorrel, M. Eichhorn, S. ed’Astorg, A. Leriche.

Please correct the author names in ref. [18]. They should be written as V.B. Glushkova, V.A. Krzhizhanovskaya, O.N. Egorova, Yu.P. Udalov, L.P. Kachalova.

Please correct ref. [19]. It should be written as W.D. Kingery, H.K. Bowen, D.R. Uhlmann. Introduction to ceramic. USA: A Wiley Interscience Press, 1975.

Please check ref. [20]. It seems to be incomplete.

You cite the same paper twice, as ref. [2] and ref. [9]. Please delete ref. [9] and change reference numbers throughout the manuscript.

Reviewer 4 Report

The presented results are original and valuable, from scientific and application points of view. A few minor questions/notes should be made clear before the manuscript acceptation.

General note.

1.    The transmission of Yb:YAG ceramics is less than the theoretical level. What are the reasons? What are the opportunities to improve the result?

2. As is known, the method of homogeneous precipitation makes it possible to vary the diameter of spheres over a wide range. Why the size of Y2O3 - 200 nm and Al2O3 - 400 nm was chosen for ceramics obtaining?

3. Conclusions need to be enriched with concrete results.

Other notes.

1. The word "home-made" in "home-made monodispersed spherical Y2O3 and Al2O3 powders" in the Сonclusion is not appropriate and should be replaced.

2.    Pay attention to the column Acknowledgments and Data Availability Statement.

Round 2

Reviewer 1 Report

The authors have significantly improved the manuscript, reacting constructively to all the comments of the reviewers. So, the paper can be recommended for publication.

Author Response

The authors have significantly improved the manuscript, reacting constructively to all the comments of the reviewers. So, the paper can be recommended for publication.

Answer: Thank you for your affirmation of our work.

Reviewer 2 Report

The authors took into account all the comments / recommendations of the reviewer, thus significantly improving the manuscript, which can now undoubtedly be recommended for publication.

Author Response

The authors took into account all the comments / recommendations of the reviewer, thus significantly improving the manuscript, which can now undoubtedly be recommended for publication.

Answer: Thank you for your affirmation of our work.

Reviewer 3 Report

The manuscript requires minor revision.

Fig. 1 (c) is absent. Please add this figure to the manuscript.

Lines 253 – 257. In Fig. 9, I cannot find the absorption peak at 300 nm.

Please explain the origin of the peak at 250 nm, which remains after annealing.

In your response to my comment “… authors state that “it is thus easier for Yb3+ to enter the YAG lattice.” Please estimate the lattice parameter of Yb-doped YAG crystals for x = 0, 0.01 and 0.10 and discuss its change with Yb concentration, if any.”, you gave an important information: “After calculation, the lattice parameter of the (Y1-xYbx)AG (x=0, 0.01, 0.10) crystals are 1.2002 nm, 1.1986 nm and 1.1964 nm respectively. Because the ionic radius of Yb is smaller than that of Y ionic radius. After Yb entered the YAG lattice, it will lead to lattice contraction. So the lattice constant decreased with the increase of Yb doping.” I advise to add these data to the text of the manuscript.

In your response to my next comment, “Please check if the Beer’s law is fulfilled for the absorption band of Er3+.”, you wrote, “Mathematical expression of Beer-Lambert law: A=lg(1/T)=Kbc. T is the transmission ratio (light transmittance), B is the absorption layer thickness. Obviously, the absorption band of Er:YAG transparent ceramic does not satisfy Beer's law”. This statement is very important. It should be placed into the text of the manuscript. The origin of this behavior should be discussed.

Lines 32-34. With the added new sentence, the text requires correction because Ce3+, Eu3+ are also rare earth ions. It can be corrected as follows, “YAG doped with (Ce3+, Eu3+) can be widely used in the fields of scintillation and luminescent materials [7]. YAG doped with other rare earth ions (Nd3+, Yb3+) can be widely used as laser crystal in the fields of laser, medical instruments, and military defense.”

Please correct the author names in ref. [3]. I am sorry in my previous review I did not correct all the family names. They should be written as F.F. Malyavin, V.A. Tarala, S.V. Kuznetsov, A.A. Kravtsov, I.S. Chikulina, M.S. Shama, E.V. Medyanik, V.S. Ziryanov, E.A. Evtushenko, D.S. Vakalov, V.A. Lapin, D.S. Kuleshov, L.V. Tarala, L.M. Mitrofanenko.

Lines 48 – 50. The sentence requires grammar correction.

Lines 255 – 257. The sentence requires grammar correction.
